# Tensor-network approach for quantum metrology in many-body quantum systems

Krzysztof Chabuda[1], Jacek Dziarmaga[2], Tobias J. Osborne[3] & Rafał Demkowicz-Dobrzański [1]*

Identification of the optimal quantum metrological protocols in realistic many particle quantum models is in general a challenge that cannot be efficiently addressed by the state-of-the-art numerical and analytical methods. Here we provide a comprehensive framework exploiting matrix product operators (MPO) type tensor networks for quantum metrological problems. The maximal achievable estimation precision as well as the optimal probe states in previously inaccessible regimes can be identified including models with short-range noise correlations. Moreover, the application of infinite MPO (iMPO) techniques allows for a direct and efficient determination of the asymptotic precision in the limit of infinite particle numbers. We illustrate the potential of our framework in terms of an atomic clock stabilization (temporal noise correlation) example as well as magnetic field sensing (spatial noise correlations). As a byproduct, the developed methods may be used to calculate the fidelity susceptibility—a parameter widely used to study phase transitions.

[1] Faculty of Physics, University of Warsaw, ul. Pasteura 5, 02-093 Warszawa, Poland. [2] Institute of Physics, Jagiellonian University, Łojasiewicza 11, 30-348 Kraków, Poland. [3] Institut für Theoretische Physik, Leibniz Universität Hannover, Appelstraße 2, 30167 Hannover, Germany. *email: demko@fuw.edu.pl

Quantum metrology[1–6] is plagued by the same computational difficulties afflicting all quantum information processing technologies, namely, the exponential growth of the dimension of many particle Hilbert space[7,8]. Apart from idealized noiseless models[1] as well as models operating within the fully symmetric subspace[9] (where the Hilbert space dimension grows linearly with the number of particles) only small-scale problems are feasible via direct numerical study, and even a slight increase in the number of elementary objects makes such an approach intractable. Interestingly, in case of uncorrelated noise models, easily computable fundamental precision bounds are available[10–16] and hence a deep-physical insight may be obtained even if direct numerical optimization is infeasible. However, in cases when one deals with metrological models involving correlated noise, or whenever states outside the fully symmetric subspace are involved, there are no efficient methods that can be applied.

Correlated noise appears naturally in a number of highly relevant metrological problems. Temporal noise correlations are present in the atomic clock stabilization problem[17], making identification of the optimal quantum clock stabilization strategies a highly non-trivial task[18–20]. An even more challenging case involves models where time-correlated noise cannot be effectively described via some classical stochastic process[21] and as such manifests non-Markovian features of quantum dynamics[22], as, e.g., in NV-center sensing models[23]. A second natural setting exhibiting non-trivial noise correlations is that of many-body systems such as, e.g., spin chains. Here, typically, spatially correlated noise emerges, which is of crucial relevance for any models where the effective signal is obtained from spatially distributed probes[24–26].

Temporal noise correlations usually decay rapidly. Similarly, in the spatially correlated case one expects on dimensional and energetic grounds that noise will be short-range correlated. The most successful approach for classical simulating short-range correlated many-body systems is via the variational tensor-network state (TNS) ansatz[27]. Among many ansatz classes, that led to unparalleled insights into the physics of quantum many-body systems, the most relevant for the present work is the matrix product operator (MPO) ansatz for density operators[28] and also its infinite particle limit known as infinite MPO (iMPO)[29].

In this paper, building upon experience obtained from uncorrelated noise metrological studies[30], where the optimal input states where shown to be efficiently described as matrix product states (MPS), we develop a comprehensive tensor-network-based framework allowing to (i) calculate relevant metrological quantities (such as the Quantum Fisher Information (QFI) or a Bayesian-type cost), (ii) optimize input probe states and as a result (iii) identify the optimal metrological protocol. All this is accomplished while remaining fully within the tensor-network formalism and thus avoiding the curse of dimensionality along the way.

## Results

### Efficient identification of the optimal quantum metrological protocol.
A paradigmatic task in quantum metrology is schematically depicted in Fig. 1. The central goal is to find the best input probe state $\rho_0$, the best measurement and estimator so as to minimize the average uncertainty $\Delta^2 \widetilde{\varphi} = \langle (\widetilde{\varphi} - \varphi)^2 \rangle$, where the expectation is over all measurement results $x$.

This task is somewhat facilitated by exploiting a fundamental result in quantum metrology, namely, the Cramér-Rao inequality[31,32], which lower-bounds the average uncertainty of the best possible estimator $\widetilde{\varphi}(x)$

$$\Delta^2 \widetilde{\varphi} \geq \frac{1}{F(\rho_\varphi)}, \tag{1}$$

where $F(\rho_\varphi)$ is the QFI of the output state.

**Fig. 1 A scheme of a standard quantum metrological task.** A probe state $\rho_0$ is subject to a parameter-dependent quantum evolution, mathematically represented by a parameter-dependent quantum channel $\Lambda_\varphi$. A POVM type measurement $\{\Pi_x\}_x$ is then carried out yielding a conditional probability distribution $p(x|\varphi) = \mathrm{Tr}(\rho_\varphi \Pi_x)$, where $\rho_\varphi = \Lambda_\varphi(\rho_0)$. Given the conditional probability distribution $p(x|\varphi)$ the objective is to estimate the value of the unknown parameter $\varphi$. To this end one employs an estimator function $\widetilde{\varphi}(x)$ to produce a given estimate for $\varphi$ given the measurement outcome $x$.

In this paper, we will use the following formula for the QFI[9,33]:

$$F(\rho_\varphi) = \sup_L F(\rho_\varphi, L), \quad F(\rho_\varphi, L) = 2\mathrm{Tr}\left(\rho'_\varphi L\right) - \mathrm{Tr}\left(\rho_\varphi L^2\right), \tag{2}$$

where $\rho'_\varphi$ is the derivative of $\rho_\varphi$ with respect to $\varphi$. This form is equivalent to the standard definition of the QFI[31,32], as can be seen by solving the above maximization problem with respect to $L$—this is formally a quadratic function in matrix $L$ and the resulting extremum condition yields the standard linear equation for the symmetric logarithmic derivative (SLD) $L$, $\rho'_\varphi = \frac{1}{2}(L\rho_\varphi + \rho_\varphi L)$. When the solution for $L$ is plugged into the above formula it yields $F(\rho_\varphi) = \mathrm{Tr}(\rho_\varphi L^2)$ in agreement with the standard definition of the QFI.

The above QFI formula has an advantage over the standard one, when one wants to additionally perform optimization of the QFI over the input states $\rho_0$ in order to find the optimal quantum metrological protocol. This problem can be written as a double maximization problem:

$$F = \sup_{\rho_0} F[\Lambda_\varphi(\rho_0)] = \sup_{\rho_0, L} F[\Lambda_\varphi(\rho_0), L]. \tag{3}$$

With $\Lambda_\varphi$ fixed $F[\Lambda_\varphi(\rho_0)]$ is effectively a function of $\rho_0$, so in what follows for the sake of simplicity of notation we will write $F(\rho_0)$, and $F(\rho_0, L)$ instead of $F(\rho_\varphi)$, $F(\rho_\varphi, L)$. The Figure of Merit (FoM) $F(\rho_0, L)$ is linear in $\rho_0$ and quadratic in $L$. This formulation leads to an extremely efficient iterative numerical procedure for determining the optimal input probe state: start with some random (or an educated guess for an) input state, determine the corresponding optimal $L$ by performing the relevant optimization. Then, for the $L$ just found, reverse the procedure and look for the optimal input state. This procedure converges very quickly and yields the optimal input probe state, as well as the corresponding QFI. This approach was first proposed in refs. [9,34] in the Bayesian estimation context, and then applied to the QFI FoM in ref. [33] (recently the method has been rediscovered in a slightly modified incarnation in ref. [35] and proved useful in studying metrological properties of states with a positive partial transpose). The above considerations are valid whenever the quantity to be optimized is given in the form of Eq. (2): $\rho'_\varphi$ need not necessarily be the derivative of the state with respect to the estimated parameter. As such, this procedure is applicable in the Bayesian approach, and also in case of less trivial FoMs (see the atomic clock stabilization example).

We consider metrological models involving $N$ distinguishable $d$-dimensional probes, so that the total Hilbert space is $\mathcal{H} = \bigotimes_{n=1}^{N} \mathbb{C}^d$. We assume that the parameter $\varphi$ is unitarily encoded in the output state $\rho_\varphi$ according to a product of unitaries given by the exponential of local generators (or Hamiltonians):

$$\rho_\varphi = \Lambda_\varphi(\rho_0) = e^{-iH\varphi} \Lambda(\rho_0) e^{iH\varphi}, \quad H = \sum_{n=1}^{N} h^{[n]}, \tag{4}$$

where $h^{[n]}$ is the generator acting on the $n$th particle. Most importantly for this paper, the noise, represented above by the operator $\Lambda$, is not assumed to be local, but admits the possibility of noise correlations. In order to be able to make use of MPO approach efficiently we assume that the noise correlations are short-range, i.e., $\Lambda$ may be effectively approximated as a product of single, two-, three-, etc. particle maps up to some cutoff point after which to is assumed that noise correlations do not extend beyond $r$ neighboring particles—see Supplementary Note 1 for a more detailed and more physical discussion of this approximation.

**Expressing the optimization problem using the MPO formalism.** The optimization scheme described above is completely general and, provided the systems considered are small enough, it can be implemented numerically using the standard quantum state representation. For larger systems, however, this will not be possible in general, and one way to circumvent this difficulty is to implement the above algorithm using the MPO formalism.

In order to do so, we first write the initial $N$ particle state $\rho_0$ as an MPO (see Supplementary Note 4 for a short introduction to tensor networks):

$$\rho_0 = \sum_{\mathbf{j},\mathbf{k}} \mathrm{Tr}\left( R_0[1]_{k_1}^{j_1} R_0[2]_{k_2}^{j_2} \dots R_0[N]_{k_N}^{j_N} \right) |\mathbf{j}\rangle\langle\mathbf{k}|, \quad (5)$$

where $|\mathbf{j}\rangle = |j_1, \dots, j_N\rangle$ denotes the standard product basis for $N$-particle states, with $j_n = 0, \dots, d-1$ being the physical indices, while $R_0[n]_{k_n}^{j_n}$ are $D_{\rho_0} \times D_{\rho_0}$ matrices (for which entries are identified by virtual indices $\alpha, \beta = 1, \dots, D_{\rho_0}$) and $D_{\rho_0}$ is referred to as the bond-dimension of the MPO. The above state is depicted diagrammatically below

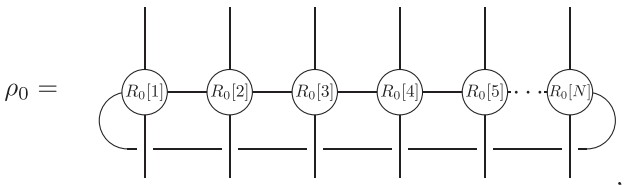

where the contracted lines represent virtual indices and uncontracted lines physical indices. We may vectorize $\rho_0$ and obtain its MPS representation by formally bending the vertical legs upward ($|j\rangle\langle k| \to |j, k\rangle$), which results in

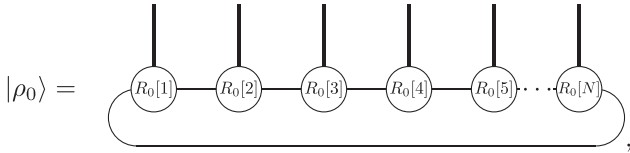

where a thick vertical line $\mathbf{|} = \mathbf{||}$ ranges over a doubled physical index $(j, k)$.

Our next step is to find the tensor-network representation of $\rho_\varphi = \Lambda_\varphi(\rho_0)$. For definiteness, we focus on the situation when the noise operator $\Lambda$ can be described by a subsequent action of single-particle $\Lambda^{[n]}$ and two-particle terms $\Lambda^{[n,n+1]}$—which in the following are denoted by $Y$ and $X$, respectively. Physically this is the case when single and two-particle evolution terms commute and no particle is distinguished—generalizations to more complex situations are tedious but straightforward. In the tensor-network representation the action of the $\Lambda$ operator takes the

following form:

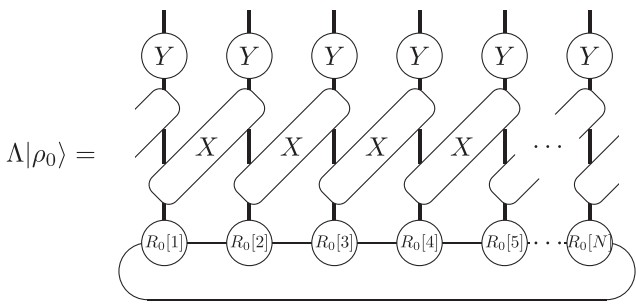

where we place $X$ operators in a skewed orientation in order to maintain a manifestly translation invariant model.

The operator $X$ is defined with respect to a product basis $|\alpha\rangle = |j, k\rangle$ for the doubled legs and it acts on two neighboring subsystems $|\alpha\rangle$ and $|\beta\rangle$, i.e., it is the tensor $X = \sum_{\alpha',\alpha,\beta',\beta} X_{\alpha',\alpha,\beta',\beta} |\alpha'\rangle\langle\alpha| \otimes |\beta'\rangle\langle\beta|$. We may now perform the singular value decomposition (SVD) of tensor $X$, which has the following graphical representation:

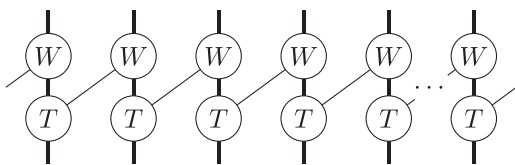

Here, we regard the two vertical legs of $X$ on the left as a doubled leg acting on a single virtual system and the two vertical legs on the right as acting on a second virtual system: in this way one can think of $X$ as a simple matrix acting on a virtual system and we can then apply the SVD.

We apply the SVD to each $X$ operator and absorb $\sqrt{S}$ into the $U$ tensor from the right (respectively, into the $V^\dagger$ tensor from the left). Let $D^{(2)}$ (the upper index indicates two-particle nature of the noise) be the number of non-zero (or more practically non-negligible) singular values $s_\gamma$ of matrix $S$. Introduction of $T = U\sqrt{S}$, $W = \sqrt{S}V^\dagger$ finalizes the decomposition of the $X$-layer of the tensor-network:

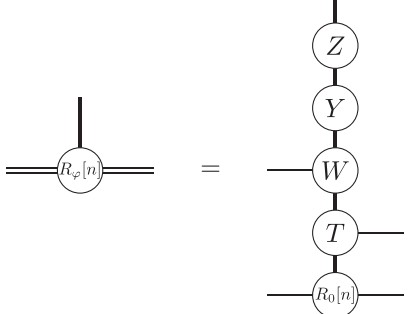

The final step to obtain an MPO representation for $\rho_\varphi$ is to combine the tensors $R_0[n]$, $T$, $W$, $Y$, and a tensor $Z = e^{-ih\varphi} \otimes (e^{ih\varphi})^{\mathrm{T}}$—which represents the unitary phase encoding process—into a single new MPS tensor $R_\varphi[n]$:

The doubled horizontal legs in the scheme above can be now combined into thicker horizontal legs to obtain the MPO representation of $\rho_\varphi$ (note that we have split back the vertical legs so we have a proper density matrix, and not its vectorized

form):

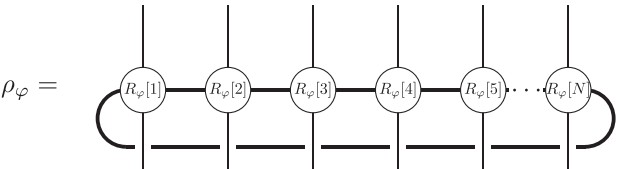

As a result we end up with an MPO with the bond-dimension $D_\rho = D_{\rho_0} D^{(2)}$. The generalization of this derivation beyond the case of nearest-neighbor correlations will lead to an MPO representation of the density matrix $\rho_\varphi$ with bond-dimension $D_\rho = D_{\rho_0} D_r$. Here, $D_r = \prod_{s=2}^r D^{(s)}$ represents the contribution to the effective bond-dimension of the output state resulting from the action of the correlated noise, where $D^{(s)}$ is the number of non-zero singular values that will appear when considering the $s$-particle noise term.

Operator $\rho'_\varphi$ can be efficiently written as an MPO as well thanks to the fact that it is as a commutator of $\rho_\varphi$ with $H$, where $H$ is a sum of local Hamiltonians

$$\rho'_\varphi = \sum_{n=1}^{N} \mathbf{i}\left[\rho_\varphi, h^{[n]}\right]. \tag{6}$$

In what follows we assume that the basis $|j\rangle$ associated with the physical indices $j$ is chosen to be the eigenbasis of the local Hamiltonian $h$, $h = \sum_j \epsilon_j |j\rangle\langle j|$, where $\epsilon_j$ are the corresponding eigenvalues. The MPO representation of $\rho'_\varphi$ can be written at the cost of doubling the bond-dimension ($D_{\rho'} = 2D_{\rho_0} D_r$):

$$\rho'_\varphi = \sum_{\mathbf{j},\mathbf{k}} \mathrm{Tr}\left(R'[1]_{k_1}^{j_1} R'[2]_{k_2}^{j_2} \dots R'[N]_{k_N}^{j_N}\right)|\mathbf{j}\rangle\langle\mathbf{k}|, \tag{7}$$

where $R'[n]_{k_n}^{j_n}$ is equal to

$$\begin{cases} \begin{pmatrix} \mathbf{i}(\epsilon_{k_1} - \epsilon_{j_1}) & 1 \\ 0 & 0 \end{pmatrix} \otimes R_\varphi[1]_{k_1}^{j_1} & \text{for } n = 1, \\[12pt] \begin{pmatrix} 1 & 0 \\ \mathbf{i}(\epsilon_{k_n} - \epsilon_{j_n}) & 1 \end{pmatrix} \otimes R_\varphi[n]_{k_n}^{j_n} & \text{for } n \in [2, \dots, N-1], \\[12pt] \begin{pmatrix} 1 & 0 \\ \mathbf{i}(\epsilon_{k_N} - \epsilon_{j_N}) & 0 \end{pmatrix} \otimes R_\varphi[N]_{k_N}^{j_N} & \text{for } n = N. \end{cases} \tag{8}$$

The $2 \times 2$ matrices that appear in the above construction are responsible for the increase of the bond-dimension but guarantee that the effect of trace in Eq. (7) is equivalent to that resulting from the sum of $\rho_\varphi$ MPO acted upon consecutively by the commutator of the local Hamiltonians corresponding to different particles.

Finally, the optimization algorithm needs to be implemented using the MPO structures introduced above. Here, we just sketch how the two-step iterative process of identifying the optimal $L$ and $\rho_0$ is implemented within the MPO framework in order to arrive at the solution of Eq. (3)—see the Methods section for a detailed description. In the first iteration step we seek the maximum of the $F(\rho_0, L)$ over a Hermitian operator $L$ with a fixed initial density matrix $\rho_0$ (QFI is a quadratic function of $L$). We use the MPO representation of $L$ and, instead of trying to optimize the whole operator $L$ at once, we iteratively optimize each tensor site by site in a loop up to the desired convergence of the FoM. During the one-site optimization step we express our FoM as a quadratic function

of the vectorized tensor, which we want to optimize. This gives us a condition for the extremum in a form of a linear equation for the components of this tensor, which we solve using the Moore–Penrose pseudo-inverse. The second iteration step, where the optimization over the initial state is performed, is similar in concept but varies in details. One can show that the optimal input state is pure and then using the Heisenberg picture of the evolution process express the optimization problem as a minimization of a Rayleight quotient involving a Hermitian operator and the respective input state. When the Hermitian operator is interpreted as a Hamiltonian then such a problem reduces to the well known variational method in quantum mechanics, which when the MPS formalism is incorporated can be efficiently solved using the DMRG algorithm[27].

Note that in the above process we have an outer iteration loop related with the iteration process where we subsequently optimize $L$ and $\rho_0$ in the FoM quantity $F(\rho_0, L)$ and at the same time we have an inner iteration loop in each of these steps where we iterate over sites of the relevant tensor-network in order to find the optimal solution within a given class of MPO. Furthermore, while running the algorithm, one has to choose the bond-dimensions for the input state, as well as for the $L$ operator. We start with low bond-dimensions (typically 1 or 2) and then increase them subsequently only when the increase yields a noticable change (>1%) in the optimized FoM. In this way we achieve the desired accuracy while keeping the bond-dimensions as low as possible thus guaranteeing the efficiency of the algorithm. This procedure may be implemented both in the finite particle number regime as well in the asymptotic limit of infinite number of particles by utilizing the iMPO—see the Methods section. This latter approach provides us with a unique insight into the asymptotic efficiency of the quantum enhanced metrological protocols.

Below we present three applications of our framework chosen in a way so as to highlight the possibility of applying the framework to a variety of qualitatively different physical problems and therefore demonstrate versatility of the approach.

**Magnetic field sensing with locally correlated noise.** Consider $N$ particles each with spin-$\frac{1}{2}$, which interact for a fixed time $t$ with an external magnetic field $B$ (assumed to be in the $z$ direction) whose strength fluctuates. The fluctuations induce an effective dephasing process on the particles and apart from the uncorrelated dephasing contribution[10,36] we will also take into account correlations between field fluctuations at the nearest-neighbor particle sites. The magnetic field at site $n$ is given as $B^{[n]}(t) = B + \delta B^{[n]}(t)$, where $B$ is the mean value of the field to be estimated. Here, we assume that fluctuations are Gaussian and have no relevant temporal correlations. The corresponding variance as well as the nearest-neighbor correlation functions of the fluctuating field read: $\langle \delta B^{[n]}(t)\delta B^{[n]}(t')\rangle = \sigma^2 \delta(t - t')$, $\langle \delta B^{[n]}(t)\delta B^{[n+1]}(t')\rangle = \chi \delta(t - t')$, where $\chi$ represents the strength of correlations and may be both positive and negative (anti-correlation)—for simplicity, we assume periodic boundary conditions, so in fact also the particles $N$ and 1 are correlated. This model corresponds to the choice $h^{[n]} = \sigma_z^{[n]}/2$, $\varphi = gBt$ (where $g$ is the gyromagnetic ratio for the particle), whereas using the standard cumulant expansion techniques the field fluctuations lead to:

$$\Lambda(\rho_0) = \sum_{\mathbf{j},\mathbf{k}} \langle \mathbf{j}|\rho_0|\mathbf{k}\rangle e^{-\frac{1}{2}(\mathbf{j}-\mathbf{k})^{\mathrm{T}} C(\mathbf{j}-\mathbf{k})} |\mathbf{j}\rangle\langle\mathbf{k}|, \tag{9}$$

where $\mathbf{j}, \mathbf{k}$ are column vectors $\mathbf{j} = (j_1, j_2, \dots, j_N)^{\mathrm{T}}$ ($j_n = \pm\frac{1}{2}$), and

$C$ is the correlation matrix

$$C = \begin{pmatrix} c_1 c_2 0 \dots \\ c_2 c_2 c_1 c_2 0 c_2 c_1 \vdots \\ \ddots \vdots c_2 \dots c_1 \end{pmatrix}, \tag{10}$$

where $c_1 = \sigma^2 g^2 t$, $c_2 = \chi g^2 t$ represent, respectively, local and correlated dephasing strength—see Supplementary Note 2 for extended discussion.

In order to write the evolution manifestly in the MPO formalism, we will replace $|\mathbf{j}\rangle\langle\mathbf{k}| \rightarrow |\mathbf{j}\rangle|\mathbf{k}\rangle$, which forms a basis for the vectorized input density matrix $|\rho_0\rangle$. The action of $\Lambda$ on $\rho_0$ is identical to the action of the operator $e^\Gamma$ on $|\rho_0\rangle$, i.e., $|\Lambda(\rho_0)\rangle = e^\Gamma|\rho_0\rangle$, where

$$\Gamma = -\frac{c_1}{2}\sum_{n=1}^{N}\Upsilon^{[n]} - c_2\sum_{n=1}^{N}\Xi^{[n,n+1]}, \tag{11}$$

with

$$\Upsilon^{[n]} = \left(h^{[n]}\otimes\mathbb{1} - \mathbb{1}\otimes h^{[n]}\right)^2, \tag{12}$$

and

$$\Xi^{[n,n+1]} = \left(h^{[n]}\otimes\mathbb{1} - \mathbb{1}\otimes h^{[n]}\right)\left(h^{[n+1]}\otimes\mathbb{1} - \mathbb{1}\otimes h^{[n+1]}\right). \tag{13}$$

Note that the $\Upsilon^{[n]}$ and $\Xi^{[n,n+1]}$ mutually commute with each other, so that

$$e^\Gamma = \prod_{n=1}^{N} e^{-\frac{c_1}{2}\Upsilon^{[n]}} e^{-c_2\Xi^{[n,n+1]}}. \tag{14}$$

Denoting $Y^{[n]} = e^{-\frac{c_1}{2}\Upsilon^{[n]}}$ and $X^{[n,n+1]} = e^{-c_2\Xi^{[n,n+1]}}$ we finally obtain

$$e^\Gamma = \prod_{n=1}^{N} Y^{[n]} X^{[n,n+1]}, \tag{15}$$

which is the form of evolution the same as discussed in Result section guaranteeing efficient MPO description.

After evolution through quantum channel $\Lambda$, the phase is imprinted in our state through unitary evolution according to Eq. (4) with local Hamiltonians $h^{[n]}$—in the tensor-network picture this is represented by the action of $\prod_{n=1}^{N}Z^{[n]}$, where $Z^{[n]} = e^{-ih^{[n]}\varphi}\otimes(e^{ih^{[n]}\varphi})^{\mathrm{T}}$. Written in the basis $|\mathbf{j}\rangle$, $\rho'_\varphi = i[\rho_\varphi, H]$ reads:

$$\rho'_\varphi = \sum_{\mathbf{j},\mathbf{k}}\langle\mathbf{j}|\rho_\varphi|\mathbf{k}\rangle i\sum_n(k_n - j_n)|\mathbf{j}\rangle\langle\mathbf{k}|. \tag{16}$$

Figure 2(a) presents a comparison of results of the QFI optimization procedure for exemplary noise parameters obtained using the finite number of particles $N$ MPO approach and the asymptotic value of the QFI per particle obtained using the iMPO approach. The results obtained via the two approaches are in very good agreement. This is a numerical confirmation that indeed the iMPO approach, which, as described in the Methods section is much more conceptually involved, yields correct results. In Fig. 2b we present the contour plot depicting the asymptotic value of the QFI per particle as a function of noise parameters, where bottom left inset also demonstrates that state-of-the-art methods developed with uncorrelated noise models in mind yield bounds that are far from satisfactory in case of correlated noise models. The main qualitative feature that clearly emerges is the decrease of the optimal QFI with the increase of correlated noise part parameter $c_2$. At the same time going into the anti-correlation regime (negative $c_2$) allows for a significant increase in the achievable QFI—this is to be expected based on intuition

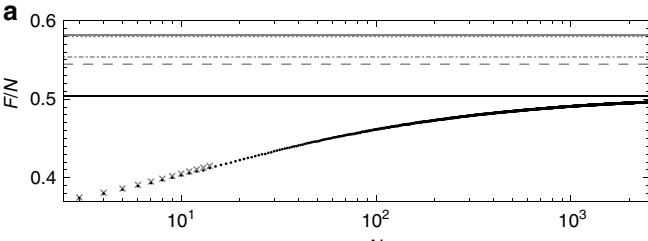

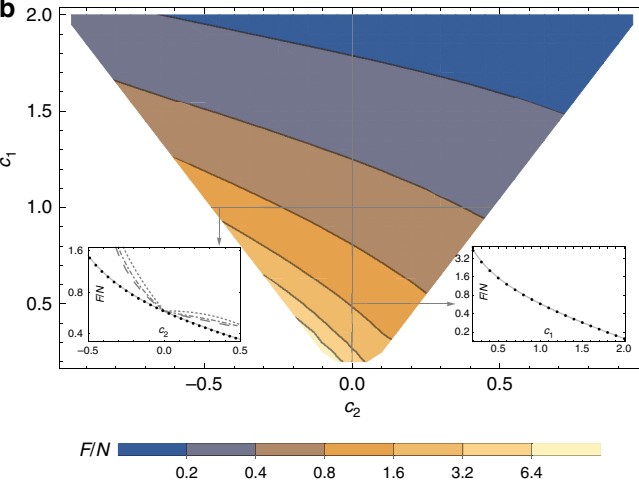

**Fig. 2 Quantum Fisher Information for the magnetic field sensing model. a** Comparison of the QFI per particle for a magnetic field sensing problem in presence of locally correlated dephasing as a function of the number of spins in a chain $N$ (for dephasing noise model with local noise parameter $c_1 = 1$ and correlation parameter $c_2 = 0.1$) calculated using the finite MPO approach (black dots) with asymptotic value obtained using the iMPO approach (black solid line). Gray crosses indicate results obtained via the standard full-Hilbert space description. Gray lines show state-of-the-art bounds on QFI/$N$ obtained by decomposing the dynamics into effectively independent channels of increasing complexity (dotted, dash-dotted, dashed)—see Supplementary Note 2 for details. For comparison, the solid gray line corresponds to the bound obtained when all correlations are neglected and only local dephasing noise is taken into account. **b** Asymptotic value (obtained using iMPOs) of QFI per particle for dephasing type noise in function of local $c_1$ and between nearest neighbors $c_2$ noise parameters. Black equipotential lines are in logarithmic scale. Left inset shows a slice of the main plot along $c_1 = 1$ and presents the results obtained using the iMPO approach (black dots) compared with the exact asymptotic result for a weakly squeezed state strategy (light gray line), and the gray lines show state-of-the-art bounds on QFI/$N$ (dotted, dash-dotted, dashed). Right inset shows a slice of the main plot along $c_2 = 0$ and presents result obtained using iMPO approach (black dots) compared with the known exact result for strictly local noise $F/N = \eta^2/(1-\eta^2)$ with $\eta = e^{-c_1/2}$ (light gray line).

obtained from quantum error-correction-based metrology where purely anti-correlated noise may be even completely removed recovering the Heisenberg scaling[26].

In case of purely local dephasing it is known that in the limit of large number of particles the fundamental bound, $F/N = \eta^2/(1-\eta^2)$, can be saturated by protocols involving weakly spin-squeezed states[10,37]. We have performed analogous analysis in case of correlated noise, see Supplementary Note 2, and found out that asymptotically the strategy involving optimally squeezed states and standard Ramsey measurement lead to the asymptotic precision $\Delta\widetilde{\varphi} = \sqrt{(1 - e^{-c_1} + 2e^{-c_1}\sinh c_2)/(Ne^{-c_1})}$, which can be related with the corresponding Fisher information

per particle equal to:

$$\frac{F}{N} = \frac{e^{-c_1}}{1 - e^{-c_1} + 2e^{-c_1}\sinh c_2}. \tag{17}$$

We have checked that this formula agrees with our numerical results up to the desired accuracy ($<1\%$), and the representative comparison of the numerical data and this formula is provided in the left inset of Fig. 2b. This implies that similarly as in the uncorrelated dephasing models, weakly spin-squeezed states are asymptotically optimal.

Using the above formula, we may also go back to the original problem of magnetic field sensing. Utilizing the relation $\varphi = gBt$ we get the corresponding magnetic field sensing precision:

$$\Delta\widetilde{B}_t = \frac{1}{gt}\Delta\widetilde{\varphi} = \frac{1}{gt}\sqrt{\frac{1 - e^{-\sigma^2 g^2 t} + 2e^{-\sigma^2 g^2 t}\sinh(\chi g^2 t)}{Ne^{-\sigma^2 g^2 t}}}. \tag{18}$$

The above formula assumes a fixed interrogation time $t$. We may generalize the considerations, and fix the total interrogation time $T$, which we allow to split into $T/t$ independent interrogation steps. The corresponding estimation uncertainty reads:

$$\Delta\widetilde{B}_T = \frac{1}{\sqrt{T/t}}\Delta\widetilde{B}_t = \sqrt{\frac{1 - e^{-\sigma^2 g^2 t} + 2e^{-\sigma^2 g^2 t}\sinh(\chi g^2 t)}{g^2 t T Ne^{-\sigma^2 g^2 t}}}, \tag{19}$$

which when optimized over $t$ reaches the minimal value when $t \to 0$ and yields:

$$\Delta\widetilde{B} = \sqrt{\frac{\sigma^2 + 2\chi}{TN}}. \tag{20}$$

Based on the above results we can expect that weakly spin-squeezed states should also be optimal in case of a more general dephasing noise, provided the range of correlations $r$ is finite and we consider the asymptotic limit $N \to \infty$. In this case, following analogous calculations, we would arrive at the optimal magnetic field sensing precision of the form

$$\Delta\widetilde{B} = \sqrt{\frac{\sigma^2 + 2\sum_{s=2}^{r}\chi_s}{TN}}, \tag{21}$$

where $\chi_s$ represent magnetic field correlations for particles at distance $s - 1$: $\langle\delta B^{[n]}(t)\delta B^{[n+s-1]}(t')\rangle = \chi_s\delta(t - t')$. Comparing this result with the performance of the GHZ states for the same model, see ref. [38], we notice that there is the $\sqrt{e}$ factor improvement in performance of the optimally spin-squeezed states over the GHZ states familiar from uncorrelated dephasing considerations[36,37].

**Atomic clock stabilization.** A typical atomic clock operates in a feedback loop where the local oscillator (LO, e.g., laser) is stabilized to atomic reference frequency by periodically interrogating atoms (using radiation from the LO) and based on the measured response, the frequency of the LO is corrected[17]. One of the main goals in the design of the clock interrogation scheme is to achieve the lowest instability typically quantified by the Allan variance (AVAR)

$$\sigma^2(\tau) = \frac{1}{2\tau^2\omega_0^2}\left\langle\left(\int_{\tau}^{2\tau}dt\omega(t) - \int_0^{\tau}dt\omega(t)\right)^2\right\rangle, \tag{22}$$

where $\langle\cdot\rangle$ represents averaging over frequency fluctuations of the LO described by some stochastic process, $\tau$ denotes averaging time, $\omega_0$ atomic reference angular frequency and $\omega(t)$ time-dependent angular frequency of the LO. The key feature from our perspective is the fact that the LO frequency fluctuations are temporally correlated and lead effectively to a temporarily correlated dephasing of atoms.

Fixing the physical properties of the atoms the goal is to optimize their initial states, interrogations times, measurements and feedback corrections in order to minimize the AVAR. Performing such a comprehensive optimization is not feasible. In

ref. [20] a lower bound on the achievable AVAR was introduced the quantum Allan Variance (QAVAR):

$$\sigma_Q^2(\tau) = \sigma_{LO}^2(\tau) - \frac{1}{\omega_0^2}\sup_{\rho_0, L, T} F_A(\tau; \rho_0, L, T), \tag{23}$$

$$F_A(\tau; \rho_0, L, T) = \left[2\mathrm{Tr}\left(\rho'L\right) - \mathrm{Tr}\left(\rho L^2\right)\right]/2, \tag{24}$$

where $\sigma_{LO}^2(\tau)$ is the AVAR of free running LO, $\frac{1}{\omega_0^2}F_A(\tau; \rho_0, L, T)$ represents a correction to it from the feedback loop and $T$ is the interrogation time. We do not provide here explicit forms of the operators $\rho$ and $\rho'$, and refer the interested reader to ref. [20], but just note that the structure of the formulas are similar to that in Eq. (2). The important information is that, if the atoms with which the atomic clock interacts are described via states on some $d$ dimensional Hilbert space $\mathcal{H}$, then the $\rho$ and $\rho'$ objects act on a tensor space $\mathcal{H}^{\otimes N}$, where $N = 2(\tau/T) - 1$ is the number of atomic cycles that need to be considered in order to calculate QAVAR.

In Fig. 3a we present the exemplary results for an atomic clock operating on one two-level atom, where the LO fluctuations are characterized by the autocorrelation function $R(t)$, which is a combination of Ornstein-Uhlenbeck (OU) process and white Gaussian frequency noise $R(t) = \alpha e^{-\gamma t} + \beta\delta(t)$ ($\alpha = 1$ (rad/s)$^2$, $\beta = 0.1$ (rad/s)$^2$s, $\gamma = 2$ s$^{-1}$) (see more detailed discussion of the model in Supplementary Note 3). In the long averaging time limit QAVAR takes the form $\sigma_Q^2(\tau) \simeq c/(\tau\omega_0^2)$ with some constant $c$, which we will refer to as asymptotic coefficient. The results indicate that completely neglecting noise correlations in the analysis of the clock performance is unjustified. These calculations have been attempted in ref. [20] using the full-Hilbert space description, but were not capable of approaching the regime where the character of the scaling of the QAVAR and the coefficient could be unambiguously read out.

Following this approach we calculate the QAVAR asymptotic coefficient $c$ and the corresponding optimal interrogation time $T$ as a function of the number of atoms in the clock, see Fig. 3b. From this figure we see that the differences in QAVAR between cases with strictly local noise and when nearest-neighbor noise correlations are included only grow with the increasing number of atoms. This implies that noise correlations play an important role in the accurate analysis of clock performance. We confront the results (which are optimized over the input state) with values obtained using a NOON/GHZ states as an input, $|\psi\rangle = (|0\rangle + |d - 1\rangle)/\sqrt{2}$. The NOON states are highly prone to dephasing noise, and hence the optimal interrogation times will be necessary reduced compared to the optimal, more robust states. This is a manifestation of a generic poor performance of the NOON/GHZ states in realistic (noisy) scenarios with increasing particle number $N$[10,11,36].

**Fidelity susceptibility for many-body thermal states.** In condensed matter context, fidelity $\mathcal{F}(\varphi, \varphi + \varepsilon) = \mathrm{Tr}\sqrt{\sqrt{\rho_\varphi}\rho_{\varphi+\varepsilon}\sqrt{\rho_\varphi}}$ between many-body states $\rho_\varphi$ and $\rho_{\varphi+\varepsilon}$, that differ by a small variation of a parameter $\varphi$ in a Hamiltonian, is a mean to identify the location $\varphi_c$ of a phase transition[41,42]. This is where the fidelity susceptibility $\chi_\varphi$, defined by $\mathcal{F}(\varphi, \varphi + \varepsilon) \approx 1 - \frac{1}{2}\chi_\varphi\varepsilon^2$, has a maximum indicating a fundamental change in the state of the system. This concept was employed in ref. [43] to evaluate the usefulness of a quantum phase transition, that happens at zero-temperature, for precise sensing of the parameter $\varphi$ in a realistic system at a finite temperature. QFI defines a metric in the space of quantum states (the Bures metric)[32] and is directly related with the fidelity susceptibility, namely $F = 4\chi_\varphi$.

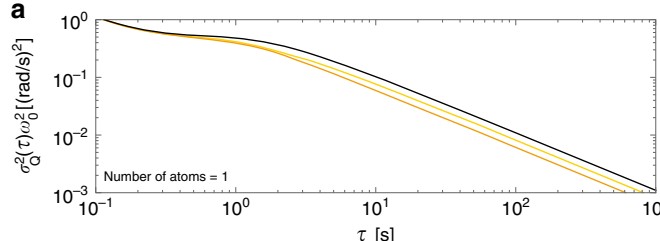

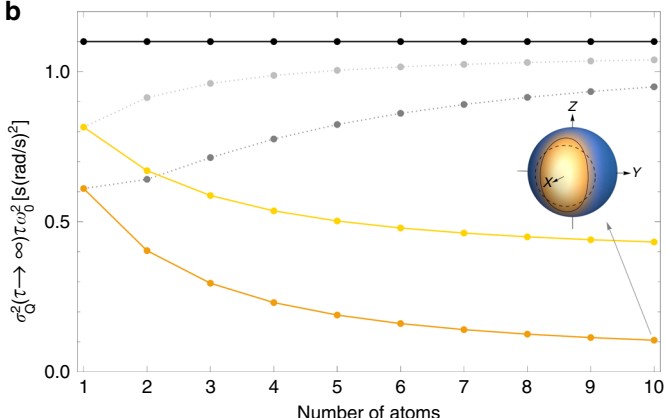

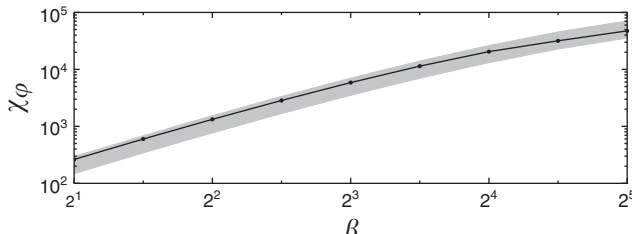

**Fig. 4 Fidelity susceptibility for the thermal state.** Exact fidelity susceptibility for a thermal many-body state (dots connected by the line) at the critical point in function of dimensionless inverse temperature $\beta$ in the XX model (Eq. 26) with 64 spins. The shaded band shows the bounds (Eq. 25). As predicted in refs. [39,40], the exact value tends to the upper/lower bound for high/low temperatures.

**Fig. 3 Quantum Allan Variance in the atomic clock stabilization model.** **a** QAVAR (times $\omega_0^2$) as a function of the averaging time $\tau$ for the atomic clock (based on one atom) with the local oscillator (LO) noise, which is strictly local (yellow line) or also includes the nearest neighbors correlations (orange line), plotted against the AVAR of uncorrected LO (black line). **b** QAVAR asymptotic coefficient as a function of the number of atoms in the atomic clock with LO noise, which is strictly local (yellow dots connected by solid line/light gray dots connected by dotted line) or also includes the nearest neighbors correlations (orange dots connected by solid line/gray dots connected by dotted line) for the optimal/NOON state, plotted against the AVAR asymptotic coefficient of uncorrected LO (black dots connected by solid line). The sphere depicts Husimi Q distribution on the Bloch sphere for the optimal state of ten atoms with marked equipotential lines where quasiprobability is equal to 0.1 for coherent spin state (black dashed line) and for this optimal state (black solid line), which shows that the state is squeezed.

Unlike at zero-temperature, the fidelity between a thermal many-body states represented by MPOs is not tractable in general. This is why a quasi-fidelity was employed $\widetilde{\mathcal{F}}(\varphi, \varphi + \varepsilon) = \sqrt{\mathrm{Tr}\sqrt{\rho_\varphi}\sqrt{\rho_{\varphi+\varepsilon}}}$ defining a quasi-susceptibility, $\widetilde{\mathcal{F}}(\varphi, \varphi + \varepsilon) \approx 1 - \frac{1}{2}\widetilde{\chi}_\varphi \varepsilon^2$, that provides bounds for the exact fidelity susceptibility[40]

$$\widetilde{\chi}_\varphi \leq \chi_\varphi \leq 2\widetilde{\chi}_\varphi. \qquad (25)$$

The Hamiltonian considered in ref. [43] was the spin-$\frac{1}{2}$ XX model

$$H = -\sum_{n=1}^{N-1}\left(\sigma_x^{[n]}\sigma_x^{[n+1]} + \sigma_y^{[n]}\sigma_y^{[n+1]}\right) + \varphi\sum_{n=1}^{N}\sigma_x^{[n]}, \qquad (26)$$

with a quantum critical point at $\varphi_c = 0$. Taking the MPOs studied in ref. [43], we bypass the tractability problem employing the part of our scheme with $\rho'_\varphi = (\rho_{\varphi+\varepsilon} - \rho_\varphi)/\varepsilon$ to calculate the QFI $F = 4\chi_\varphi$. This exact susceptibility for the chain with 64 spins is shown in Fig. 4, together with the upper and lower bounds (Eq. 25). The accuracy of the fidelity susceptibility is limited by the finite bond-dimension $D_L$ as well the finite parameter difference $\varepsilon$.

Nevertheless, we obtain satisfying results with relative error around 1% for $\varepsilon = 10^{-4}$ and $D_L = 4$.

This example demonstrates that the scheme for calculating fidelity susceptibility is a useful byproduct of our general algorithm. Beyond the present metrological context, it paves a way to generalize the zero-temperature fidelity approach to detecting quantum phase transitions[41,42]—by now standard in condensed matter physics—to phase transitions in quantum many-body systems at finite temperature.

## Discussion

We have provided a comprehensive framework for optimization of quantum metrological protocols using the tensor-network formalism. The potential to deal effectively with correlated noise models, as well as directly access the asymptotic $N \to \infty$ is what makes this framework unique. We also expect that this framework may be adapted to deal with even more challenging metrological problems including noisy multiparameter estimation[44,45], waveform estimation[46,47], or the study of the effectiveness of adaptive metrological protocols, including quantum error-correction-based schemes[16,26,48,49]. We also expect that this numerical framework may be crucial for understanding better metrological models with temporally correlated noise especially of non-Markovian nature[22], where effective tools to find the optimal metrological protocols in such cases are yet to be developed.

## Methods

We have implemented all of our algorithms in MATLAB with the help of ncon() function[50] for tensor contraction.

**Optimization in the finite number of particles regime.** As discussed in the main text maximization of our FoM $F(\rho_0, L)$, see Eq. (3), leading to the maximal possible QFI for a given metrological model is a two-step iterative process. First, we show how to maximize the FoM over a Hermitian operator $L$ with fixed $\rho_0$, and then we focus on the maximization over the input state $\rho_0$ with fixed $L$.

We search for the optimal $L$ in the form of an MPO

$$L = \sum_{\mathbf{j},\mathbf{k}} \mathrm{Tr}\left(S[1]_{k_1}^{j_1} S[2]_{k_2}^{j_2} \ldots S[N]_{k_N}^{j_N}\right)|\mathbf{j}\rangle\langle\mathbf{k}|, \qquad (27)$$

with a finite bond-dimension $D_L$. Without loss of generality, each $S[n]_{k_n}^{j_n}$ is assumed Hermitian in its physical indices,

$$S[n]_{k_n}^{j_n} = \overline{S[n]_{j_n}^{k_n}}, \qquad (28)$$

to ensure that $L$ is Hermitian. We call this condition a Hermitian gauge.

The bond-dimension $D_L$ for $L$ is expected to be small for weakly correlated noise models. Indeed, in the limit of an uncorrelated product state $\rho_\varphi = \varrho_\varphi^{\otimes N}$, $L$ is a sum of local operators, $L = \sum_{n=1}^N L^{[n]}$. Here, $L^{[n]}$ is the SLD for the single-particle problem applied to particle $n$. In a similar way as for $\rho'_\varphi$ presented in the main text,

the sum can be represented by an MPO with a bond-dimension 2. Therefore, $D_L = 2$ is the limiting value for uncorrelated noise models.

Graphical representation of the FoM $F(\rho_0, L)$ reads:

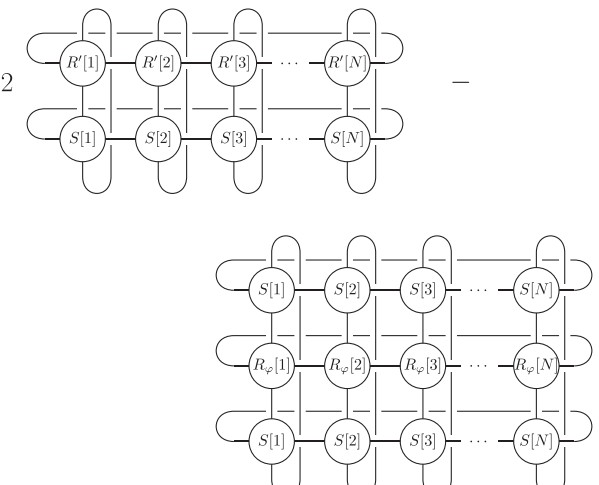

Maximization of the FoM over $L$ is equivalent to a joint maximization over each tensor $S[n]$. We relax this optimization problem by iterating an optimization loop. In the loop we first find the optimal $S[1]$, then $S[2]$, and so on up to $S[N]$, after which we go back to $S[1]$. The optimization over each $S[n]$ is performed with all other tensors fixed. The loop is repeated until the FoM converges.

After fixing the other tensors, the FoM becomes quadratic in $S[n]$ and the optimal $S[n]$ is found as a solution to a linear equation. For definiteness, to explain the procedure, we focus on the generic example of $S[2]$. We start by vectorizing $S[2] \rightarrow |S[2]\rangle$:

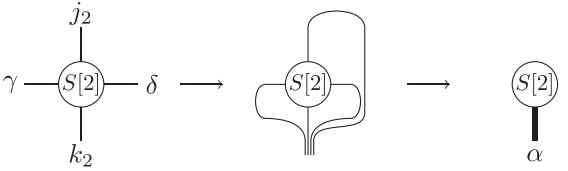

This allows us to represent the $S[2]$-FoM as:

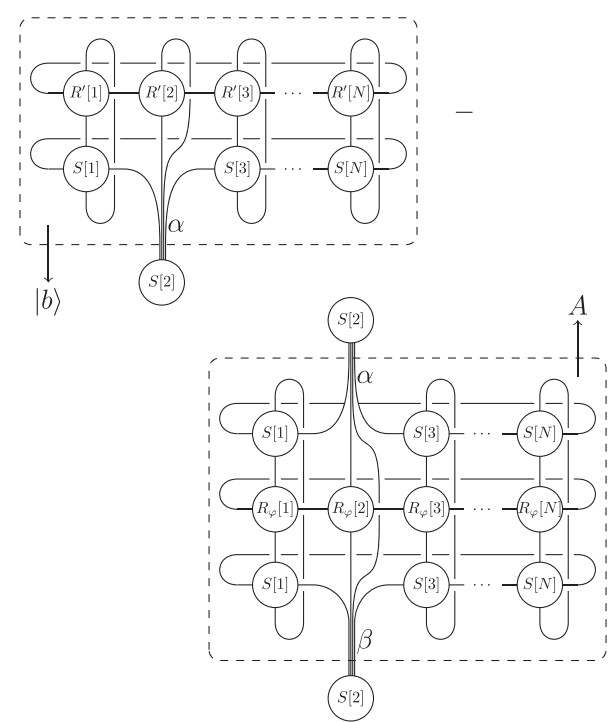

which can be also written in a compact way as

$$F(\rho_0, L) = 2 \sum_\alpha b_\alpha S[2]_\alpha - \sum_{\alpha\beta} S[2]_\alpha A_{\alpha\beta} S[2]_\beta. \tag{29}$$

Here, $b_\alpha$ are the elements of the vector $|b\rangle$, and $A_{\alpha,\beta}$ are the elements of the matrix $A$. Both $|b\rangle$ and $A$ describe the entire tensor-network complementing the distinguished vector $|S[2]\rangle$ in the two respective terms of the $S[2]$-FoM. After taking a derivative with respect to $S[2]_\alpha$, we obtain a linear equation for the extremum:

$$\frac{1}{2}(A + A^{\mathrm{T}})|S[2]\rangle = |b\rangle. \tag{30}$$

The $d^2 D_L^2 \times d^2 D_L^2$ matrix $\widetilde{A} = \frac{1}{2}(A + A^{\mathrm{T}})$ typically has a non-zero kernel and the linear equation does not have a unique solution. We use the Moore–Penrose pseudo-inverse, $\widetilde{A}^+$, to obtain a solution $|S[2]\rangle = \widetilde{A}^+|b\rangle$ that does not contain any zero modes of $\widetilde{A}$.

If the linear equation was non-singular, then its exact solution would satisfy the Hermitian gauge (Eq. 28). For the typical singular case, using an SVD of $\widetilde{A}$ to construct its pseudo-inverse, we have to truncate singular values falling below a small but finite cutoff, set by $\kappa$ multiplied by the highest singular value. As the cutoff solution $|S[2]\rangle$ need not satisfy the Hermitian gauge condition exactly, we filter out its small anti-Hermitian part with the substitution:

$$S[2]_{k_2}^{j_2} \rightarrow \frac{1}{2}\left(S[2]_{k_2}^{j_2} + \overline{S[2]_{j_2}^{k_2}}\right). \tag{31}$$

From experience, this substitution can improve numerical stability but is not necessary when all initial $S[n]$ are in the Hermitian gauge (Eq. 28) and $\kappa$ is large enough to suppress the anti-Hermitian part of the solution. However, with too large a cutoff the final optimized $L$ does not reach the maximal possible value of the QFI. Therefore, we adjust $\kappa$ to obtain the highest QFI achievable without compromising the stability.

Now we move on to the maximization of the FoM over the input state $\rho_0$ for a fixed $L$. We start by rewriting $F(\rho_0, L)$ as

$$\begin{aligned} F(\rho_0, L) &= 2\mathrm{Tr}(\rho'_\varphi L) - \mathrm{Tr}(\rho_\varphi L^2) \\ &= 2\mathrm{Tr}(\mathrm{i}[\Lambda_\varphi(\rho_0), H]L) - \mathrm{Tr}(\Lambda_\varphi(\rho_0)L^2) \\ &= 2\mathrm{Tr}(\rho_0 \mathrm{i}[H, \Lambda_\varphi^*(L)]) - \mathrm{Tr}(\rho_0 \Lambda_\varphi^*(L^2)), \end{aligned} \tag{32}$$

where by $\Lambda_\varphi^*(\cdot)$ we denote the channel, which is dual to $\Lambda_\varphi(\cdot)$ (the evolution written in the Heisenberg picture). We can rewrite this as

$$F(\rho_0, L) = \mathrm{Tr}(\rho_0(2L_\varphi'^* - L_{2,\varphi}^*)), \tag{33}$$

where we introduce $L_\varphi^* = \Lambda_\varphi^*(L)$, $L_\varphi'^* = \frac{\mathrm{d}L_\varphi^*}{\mathrm{d}\varphi} = \mathrm{i}[H, L_\varphi^*]$ and $L_{2,\varphi}^* = \Lambda_\varphi^*(L^2)$. By analogy with the construction of the MPO representation for $\rho_\varphi = \Lambda_\varphi(\rho_0)$ and $\rho_\varphi' = \mathrm{i}[\rho_\varphi, H]$, we can easily construct the MPO representation of $L_{2,\varphi}^*$ and $L_\varphi'^*$ from the known MPO form of $L$. The tensors determining the MPO form of $L_{2,\varphi}^*$ and $L_\varphi'^*$ are denoted by $S_2[n]$ and $S'[n]$, respectively, and their respective bond-dimensions are $D_{L_2} = D_L^2 D_r$ and $D_{L'} = 2D_L D_r$.

The quantity $F(\rho_0, L)$ in Eq. (33) is maximal when $\rho_0$ is a projection on the eigenvector associated with the maximal eigenvalue of the Hermitian operator $2L_\varphi'^* - L_{2,\varphi}^*$. Hence, without loss of generality, we can assume a pure input state $\rho_0 = |\psi\rangle\langle\psi|$ with $|\psi\rangle$ being an MPS with bond-dimension $D_\psi$:

$$|\psi\rangle = \sum_{\mathbf{j}} \mathrm{Tr}\left(P[1]^{j_1} P[2]^{j_2} \ldots P[N]^{j_N}\right)|\mathbf{j}\rangle. \tag{34}$$

The input state $\rho_0$ has bond-dimension $D_{\rho_0} = D_\psi^2$ and its MPO tensors are $R_0[n]_{k_n}^{j_n} = P[n]^{j_n} \otimes \overline{P[n]^{k_n}}$.

The maximization of $F(\rho_0, L)$ over the input state $|\psi\rangle$ is equivalent to the variational optimization for the ground state of a many-body "Hamiltonian" $L_{2,\varphi}^* - 2L_\varphi'^*$, a problem widely discussed in the many-body physics MPS literature[27,51,52]. After reinterpreting our problem as a variational minimization of the "energy"

$$-F(\rho_0, L) = \frac{\langle\psi|L_{2,\varphi}^* - 2L_\varphi'^*|\psi\rangle}{\langle\psi|\psi\rangle}, \tag{35}$$

we proceed iteratively in a similar way as in the case of the maximization of $F(\rho_0, L)$ over $L$.

For example, in order to find the minimum over $P[2]$, we begin by vectorizing the tensor $P[2] \rightarrow |P[2]\rangle$ and expressing the "energy" (Eq. 35) as a Rayleigh quotient

$$-F(\rho_0, L) = \frac{\langle P[2]|\mathcal{F}|P[2]\rangle}{\langle P[2]|\mathcal{N}|P[2]\rangle}. \tag{36}$$

The tensor-network form of $dD_\psi^2 \times dD_\psi^2$ matrices $\mathcal{F}$ and $\mathcal{N}$ is given by

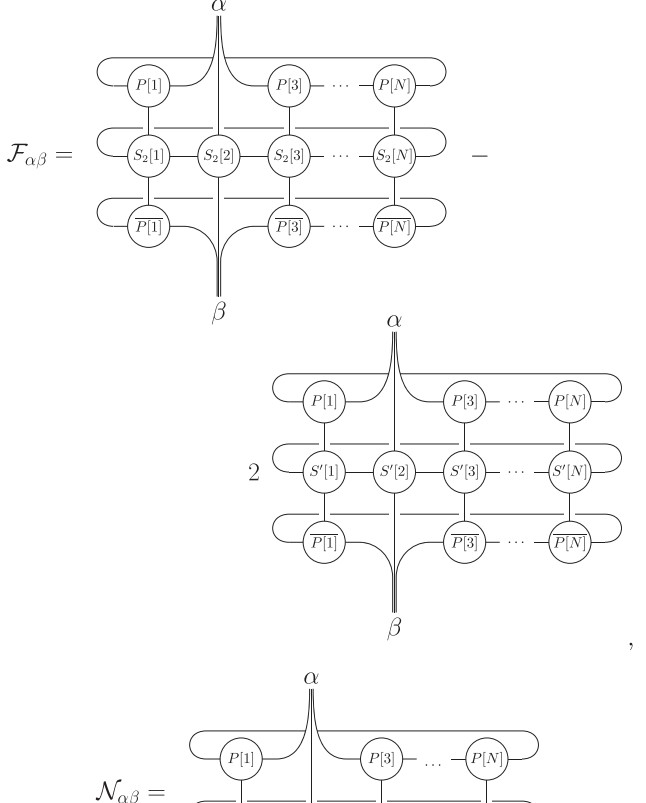

After taking the derivative of Eq. (36) we obtain the condition for the extremum:

$$\mathcal{F}|P[2]\rangle = -F(\rho_0, L)\mathcal{N}|P[2]\rangle, \tag{37}$$

which is a generalized eigenvalue problem with eigenvalue $-F(\rho_0, L)$. By multiplying it with a pseudo-inverse of matrix $\mathcal{N}$ we bring it into the form of ordinary eigenvalue problem, for which we obtain the lowest eigenvalue and its corresponding eigenvector using the Lanczos algorithm.

Modification of the entire tensor-network framework to calculate the maximal QFI for systems with open boundary conditions (OBC) poses no problem and, for systems, which are not sensitive to boundary conditions, is even advisable. In OBC the MPS representing $|\psi\rangle$ can be brought to a canonical form, where the matrix $\mathcal{N}$ becomes an identity and Eq. (37) reduces to a standard eigenvalue problem. There is no need to pseudo-invert $\mathcal{N}$.

To summarize the procedure: one needs to iteratively determine the optimal $L$ for a given $\rho_0$ and then the optimal $\rho_0$ for a given $L$, until one observes convergence of the final result, e.g., the FoM does not change more than, say, 0.1% after a fixed number of steps. From our numerical experience this happens very rapidly, typically after five iterations of the $\rho_0$ and $L$ optimization steps.

While running the algorithm, one has to choose the bond-dimensions for the input state, $D_\psi$, as well as for the SLD, $D_L$, over which the optimization is performed. As in all tensor-network algorithms, keeping the bond-dimension as low as possible is essential for their efficiency. In our calculations, we typically started with a product input state, $D_\psi = 1$, and optimized for $L$ with a minimal non-trivial $D_L = 2$. Then we increased one of the bond-dimensions, either $D_\psi$ or $D_L$, each time repeating the optimization procedure, until we found that the QFI did not change when increasing the $D$'s more than by, e.g., 1% and hence assumed that relative error of our method is around 1%.

**Optimization in the asymptotic limit**. The previous subsection described an algorithm that functions for a system with a finite number of particles $N$. For quantum metrological problems in the presence of decoherence, it is the generic situation that the optimal quantum enhancement thanks to the use of entanglement leads asymptotically (for large $N$) to an improvement by a constant factor over product-state strategies. Even though the finite-system MPO approach allows us to achieve values of $N$ that are inaccessible via exact

full-Hilbert space computation, it may sometimes be not enough to reach the asymptotic limit and determine the quantum enhancement coefficient with the desired precision. For this reason, we would like to have a procedure that allows us to go directly to the infinite particle limit, calculate the maximal achievable QFI per particle and, as a result, determine the maximal quantum enhancement coefficient.

For this purpose, we exploit the infinite MPO/MPS (iMPO/iMPS) approach, see e.g., ref. [29]. We assume that all tensors are translationally invariant (TI). Then we take the limit of infinite $N$. Technically this limit is most natural in the case of PBCs, where it is enough to notice that, for any TI transfer matrix $E$, the spectral decomposition of $E^N$ is dominated by the leading eigenvalue and eigenvector of $E$. This is why in the following discussion we proceed with PBCs. In the OBC case, which is arguably more natural in some metrological contexts, one has in principle to consider the boundary conditions at infinity. However, $E^k$ applied to a boundary vector gives the leading eigenvector of $E$ when $k$ becomes longer than the finite correlation range (just as in the Lanczos algorithm). Therefore, in the bulk (i.e., far from the boundaries), all equations the TI tensors have to satisfy become the same as for the PBC.

When the input state $|\psi\rangle$ is TI then the final state $\rho_\varphi$ is TI as well. There is a problem, however, with the operator $\rho'_\varphi$. Its construction as an MPO in Eq. (8) is not TI. Because of this, instead of calculating the derivative exactly, we approximate it by a difference of two TI iMPOs:

$$\rho'_\varphi = \frac{\rho_{\varphi+\varepsilon} - \rho_\varphi}{\varepsilon} \tag{38}$$

with infinitesimal parameter $\varepsilon$. Motivated by the defining equation for the SLD

$$\rho'_\varphi = \frac{1}{2}(L\rho_\varphi + \rho_\varphi L), \tag{39}$$

we can consider a similar to Eq. (38) expansion of the operator $L$:

$$L = \frac{\widetilde{L} - \mathbb{1}}{\varepsilon}. \tag{40}$$

Here, $\widetilde{L}$ and $\mathbb{1}$ are solutions of Eq. (39) when $\rho'_\varphi$ is replaced by, respectively, $\rho_{\varphi+\varepsilon}$ and $\rho_\varphi$. We search for the optimal $\widetilde{L}$, which is better suited for the TI formalism than the original operator $L$. Let us denote by

$$f(\rho_0, \widetilde{L}) = \frac{1}{N}F(\rho_0, L) \tag{41}$$

the QFI per particle that we want to maximize:

$$f(\rho_0, \widetilde{L}) = \frac{1}{N\varepsilon^2}\left[2\mathrm{Tr}\left(\rho_{\varphi+\varepsilon}\widetilde{L}\right) - \mathrm{Tr}\left(\rho_\varphi \widetilde{L}^2\right) - 1\right]. \tag{42}$$

A TI iMPO is defined by only one tensor, which we assign, respectively, as: $|\psi\rangle \to P$, $\rho_0 \to R_0$, $\rho_\varphi \to R_\varphi$, $\widetilde{L} \to \widetilde{S}$, $\widetilde{L}^*_\varphi \to \widetilde{S}_\varphi$, $\widetilde{L}^*_{2,\varphi} \to \widetilde{S}_2$.

Optimization of a tensor-network consisting of identical tensors $A$ is a highly nonlinear problem in $A$ and one might think that an approach similar to the one used in the previous subsection is not applicable here. Fortunately, an efficient method for the problem was developed in ref. [53]. The main idea is to find the optimal tensor $A_{\mathrm{new}}$ at one-site, treating all other tensors $A$ as fixed, and then rather then replacing all tensors by $A_{\mathrm{new}}$ perform a flexible substitution,

$$A \to A_{\mathrm{new}} \sin(\lambda\pi) - A\cos(\lambda\pi), \tag{43}$$

with a mixing angle $\lambda$. The angle is optimized to yield the best possible FoM.

We now apply this approach to our two-step iterative procedure. First we need to find the optimal $\widetilde{L}$, which is equivalent to the determination of the optimal local tensor $\widetilde{S}$.

As explained in Supplementary Note 4, the trace of an operator represented as an iMPO is defined by its transfer matrix, so we start by introducing transfer matrices $E_1$ and $E_2$ associated with, respectively, $\rho_{\varphi+\varepsilon}\widetilde{L}$ and $\rho_\varphi \widetilde{L}^2$, which are

depicted below alongside their eigendecomposition:

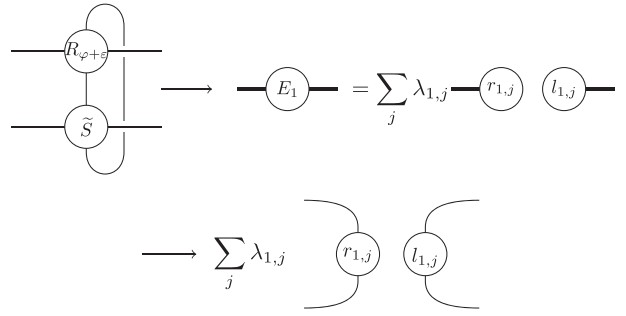

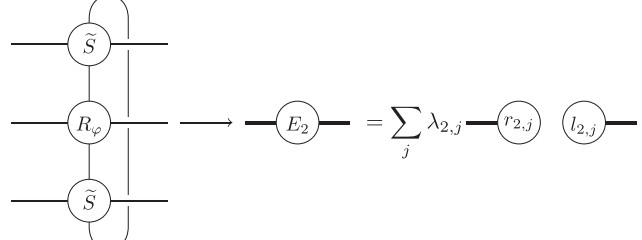

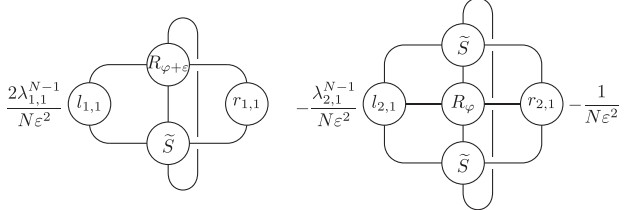

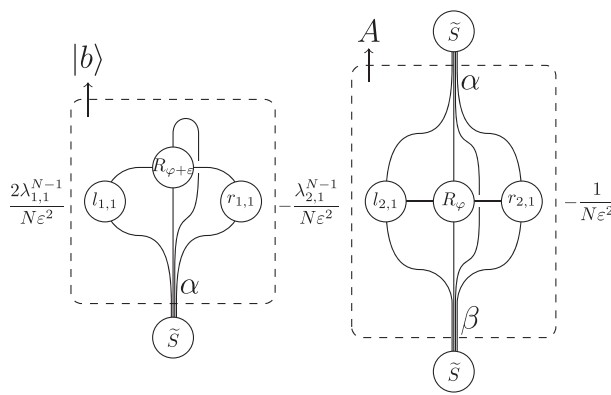

Using this transfer matrices we can write $\mathrm{Tr}\left(\rho_{\varphi+\varepsilon}\widetilde{L}\right) = \mathrm{Tr}E_1^N$, $\mathrm{Tr}\left(\rho_{\varphi}\widetilde{L}^2\right) = \mathrm{Tr}E_2^N$ and with the fact that $E_i^N$ is determined by its leading eigenvalue we can write $\mathrm{Tr}E_i^N = \lambda_{i,1}^{N-1}(l_{i,1}|E_{i,1}|r_{i,1})$. Now we can construct iMPO representation of the FoM:

where we depicted $f(\rho_0, \widetilde{L})$ and its version with tensor $\widetilde{S}$ distinguished (after performing vectorization).

This FoM can be also expressed in form of equation:

$$f(\rho_0, \widetilde{L}) = \frac{\lambda_{1,1}^{N-1}\sum_\alpha b_\alpha \widetilde{S}_\alpha - \lambda_{2,1}^{N-1}\sum_{\alpha\beta}\widetilde{S}_\alpha A_{\alpha\beta}\widetilde{S}_\beta - 1}{N\varepsilon^2}, \tag{44}$$

which give us condition for an extremum:

$$\frac{1}{2}\lambda_{1,1}^{N-1}\left(A + A^{\mathrm{T}}\right)|\widetilde{S}\rangle = \lambda_{2,1}^{N-1}|b\rangle. \tag{45}$$

For $N \to \infty$ the powers of the eigenvalues may seem to pose a problem. Fortunately, however, this problem can be circumvented. For a given $\widetilde{L}$ we can

calculate the associated value of our FoM per particle:

$$f(\rho_0, \widetilde{L}) = \frac{1}{N\varepsilon^2}(2\lambda_{1,1}^N - \lambda_{2,1}^N - 1), \tag{46}$$

but going back to Eq. (2) we see that FoM per particle should have form $f(\rho_0, \widetilde{L}) = 2f_1 - f_2$, where $f_1$ and $f_2$ are of the same order of magnitude as the asymptotic limit of the QFI per particle. Assuming that our calculations are in the regime of $N \to \infty$, $\varepsilon \to 0$, and $N\varepsilon^2 \to 0$, and remembering binomial expansion

$$(1 + \varepsilon^2 f_i)^N = 1 + N\varepsilon^2 f_i + O[(N\varepsilon^2)^2], \tag{47}$$

it is to be expected that the highest eigenvalues of the transfer matrices have the form:

$$\lambda_{1,1} = 1 + \varepsilon^2 f_1, \quad \lambda_{2,1} = 1 + \varepsilon^2 f_2, \tag{48}$$

which after inserting into Eq. (46) and using binomial expansion to the first order give us exactly $f(\rho_0, \widetilde{L}) = 2f_1 - f_2$. Note that, it means that we can calculate value of FoM per particle in a simple way:

$$f(\rho_0, \widetilde{L}) \approx \frac{1}{\varepsilon^2}(2\lambda_{1,1} - \lambda_{2,1} - 1). \tag{49}$$

It is clear now that for the purpose of solving Eq. (45) we can approximate $\lambda_{1,1}^{N-1}$ and $\lambda_{2,1}^{N-1}$ by ones and, hence, bring the condition for the optimal $\widetilde{S}$ to a simpler form:

$$\frac{1}{2}\left(A + A^{\mathrm{T}}\right)|\widetilde{S}\rangle = |b\rangle. \tag{50}$$

This equation is solved with a pseudo-inverse and its anti-Hermitian part is filtered out at every iteration step.

SLD is always traceless in any unitary parameter estimation problem, which can be seen by solving Eq. (39) for the SLD using eigenbasis of $\rho_\varphi = \sum_j \lambda_j |\lambda_j\rangle\langle\lambda_j|$:

$$L = \sum_{j,k} \frac{2\langle\lambda_j|\rho_\varphi'|\lambda_k\rangle}{\lambda_j + \lambda_k}|\lambda_j\rangle\langle\lambda_k|, \tag{51}$$

and taking into account that $\langle\lambda_j|\rho_\varphi'|\lambda_j\rangle = \mathrm{i}\langle\lambda_j|[H, \rho_\varphi]|\lambda_j\rangle = 0$. We can ensure that solution $\widetilde{S}$ has proper normalization using the condition $\mathrm{Tr}L = 0$ or, equivalently $\mathrm{Tr}\widetilde{L} = \mathrm{Tr}\mathbb{1} = d^N$, which in the language of transfer matrices means that the highest eigenvalue of the transfer matrix associated with operator $\widetilde{L}$ has to be equal to $d$. Now we turn to the second part of our optimization procedure, namely the variational minimization over the input state. This step does not introduce any qualitatively new challenges, so we only briefly discuss it for completeness. As for the $\rho_\varphi'$, we approximate the exact derivative of $L_\varphi'^*$ by its discrete version:

$$L_\varphi'^* = \frac{L_{\varphi+\varepsilon}^* - L_\varphi^*}{\varepsilon}. \tag{52}$$

After the expansion $L = (\widetilde{L} - \mathbb{1})/\varepsilon$, our task becomes equivalent to minimization of the "energy density":

$$-f(\rho_0, \widetilde{L}) = \frac{\langle\psi|\widetilde{L}_{2,\varphi}^* - 2\widetilde{L}_{\varphi+\varepsilon}^* + \mathbb{1}|\psi\rangle}{N\varepsilon^2\langle\psi|\psi\rangle} \tag{53}$$

over $|\psi\rangle$. We can depict Eq. (53) in a diagrammatic form:

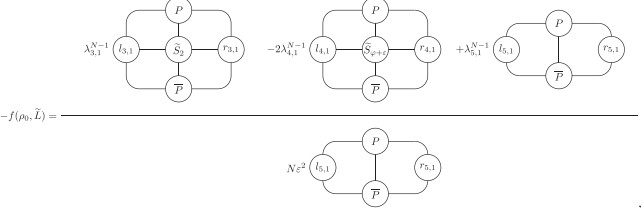

where we have used transfer matrices $E_3$, $E_4$, and $E_5$ associated with, respectively,

$\langle\psi|\widetilde{L}_{2,\varphi}^{*}|\psi\rangle$, $\langle\psi|\widetilde{L}_{\varphi+\varepsilon}^{*}|\psi\rangle$, and $\langle\psi|\psi\rangle$ that are represented graphically as

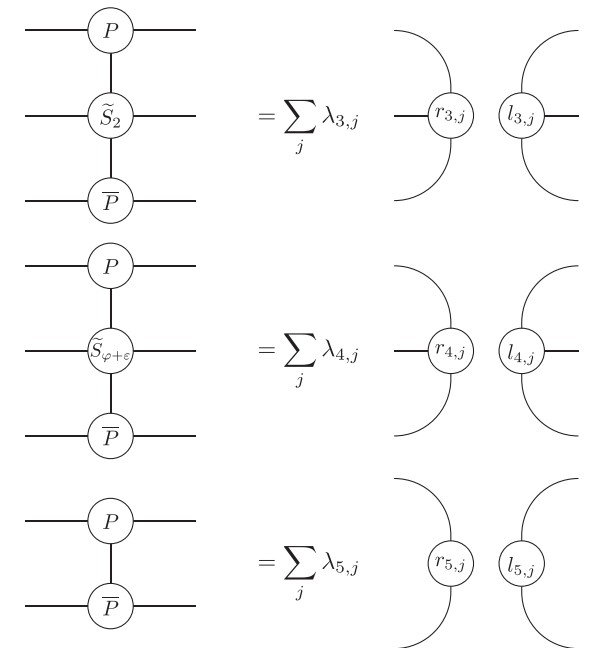

As previously, we expect that $\lambda_{i,1} = 1 + \varepsilon^2 f_i$ and for the purpose of finding the optimal tensor $P$, we can approximate $\lambda_{3,1}^{N-1}$ and $\lambda_{4,1}^{N-1}$ by ones. After taking the derivative we obtain the condition for the extremum:

$$\mathcal{F}|P\rangle = g\mathcal{N}|P\rangle, \tag{54}$$

where $g = -f(\rho_0, \widetilde{L})N\varepsilon^2 - \lambda_{5,1}^{N-1}$ is a generalized eigenvalue, whereas the matrices $\mathcal{F}$ and $\mathcal{N}$ have the following tensor-network representation

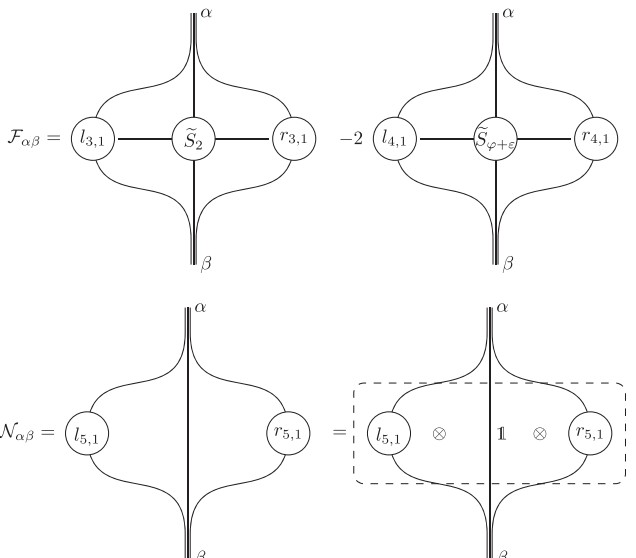

The matrix $\mathcal{N}$ is a tensor product of three matrices, $r_{5,1}$, the identity, and $l_{5,1}$, hence its pseudo-inverse $\mathcal{N}^+$ can be obtained as a tensor product of (pseudo-)inverses of the smaller matrices. Applying $\mathcal{N}^+$ to Eq. (54) we bring it into the form of a standard eigenvalue problem. We solve this eigenproblem with respect to the smallest eigenvalue and its corresponding eigenvector $|\psi\rangle$ and require that $|\psi\rangle$ is normalized so that $\lambda_{5,1} = 1$. Then we calculate the asymptotic value of the QFI per particle:

$$\begin{aligned}-f(\rho_0, \widetilde{L}) &= \frac{1}{N\varepsilon^2}(\lambda_{3,1}^N - 2\lambda_{4,1}^N + 1) \\ &\approx f_3 - 2f_4 = \frac{1}{\varepsilon^2}(\lambda_{3,1} - 2\lambda_{4,1} + 1).\end{aligned} \tag{55}$$

Just as for the finite $N$, iterating the $\widetilde{L}$ and $|\psi\rangle$ optimization steps leads to the optimal solution with the maximal QFI per particle.

While performing the numerics one should choose $\varepsilon$ to be small but not too small as too small values may lead to numerical instabilities. Our general strategy in obtaining numerical results in the examples presented, was to lower the value of $\varepsilon$ until we observed no noticeable change in the obtained results, while still remaining in the regime where algorithm was stable. In all the examples we studied in this paper this approach resulted in the choice of $\varepsilon \approx 10^{-3}$–$10^{-4}$ (instabilities started to appear for $\varepsilon < 10^{-6}$). Notice that in the asymptotic iMPO approach described above we required $N\varepsilon^2$ to be small—on the order of the precision we expect from the numerical results. In other words setting the precision requirements to $10^{-2}$ this implies that $N\varepsilon^2 \approx 10^{-2}$, and hence $N \approx 10^4 - 10^6$. What this physically means is that in our set-up the QFI per particle does not change in any noticeable way for larger $N$ and hence the asymptotic behavior in the actual $N \to \infty$ limit may be inferred from this results.

## Data availability

The data that support the findings of this study are available from the authors upon request.

## Code availability

The code used to implement the algorithms developed in the study are available from the authors upon request.

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

## Acknowledgements

We would like to thank Marek M. Rams, David Layden, Maciej Lewenstein, Shi-Ju Ran, Piet O. Schmidt, and Ian D. Leroux for fruitful discussions. We are also indebt to Marek M. Rams for sharing with us the data from ref. [43]. K.Ch. and R.D.D. acknowledge support from the National Science Center (Poland) grant No. 2016/22/E/ST2/00559. Work of J.D. was funded by the National Science Center (Poland) together with European Union through QuantERA ERA NET program 2017/25/Z/ST2/03028. T.J.O. was supported, in part, by the DFG through SFB 1227 (DQmat), the RTG 1991, and the cluster of excellence EXC 2123 QuantumFrontiers.

## Author contributions

K.Ch. played the leading role in the development and implementation of the optimization algorithms, as well as performed all numerical and part of analytical calculations. J.D. and T.J.O. provided crucial conceptual contribution to the development of the MPO algorithms as well as participated in numerous discussions. R.D.D. conceived and closely supervised the project as well as performed part of analytical calculations. All authors contributed to the preparation of the manuscript with the leading role played by K.Ch. and R.D.D.

## Competing interests

The authors declare no competing interests.
