## [Peer Review File · Nature Communications]

Reviewers' Comments:

Reviewer #1:

Remarks to the Author:

The authors present a new framework that utilises tensor networks for metrological purposes. In particular the presented method can efficiently estimate the quantum Fisher information of a given measurement scenario.

Moreover the authors provide a few important applications of this method which proves the strength of the method at hand. In my opinion this is a very important contribution that should be published in Nature Communications with some minor modifications.

My main comment is with regard to the presentation. The paper only provides the main story while all the important details are either in the method section or the supplementary material.

In my opinion some technical intuition should be added to the main text.

1) Some intuition regarding the validity of eq. 2 should be added when the equation is presented.

2) The fact that the derivative can be written as an MPO is a crucial part in the technique. This fact is mentioned in the main text but only explained in the method section. At least a short explanation should appear in the main text.

The section "Fidelity susceptibility calculations for many-body thermal states" shows that the Fisher Information is the susceptibility.

An explanation could be added how come the Fisher Information does not depend on ϵ as the derivative with respect to the susceptibility does.

The authors mention on page 5 that "This shows that any analysis of the performance of atomic clocks based on the NOON state may be mis-leading."

That fact is also known from many other investigations so should be probably erased or references should be added.

Typos:

1) page 4: states states

Reviewer #2:

Remarks to the Author:

The paper adapts known tensor network techniques to quantum metrology. The paper is quite technical and its ultimate goal is to compute the sensitivity of phase estimation protocols (namely the calculation of the quantum Fisher information) in presence of certain correlated noise (some additional assumptions are made). The applications are the standard ones — magnetic field sensing with locally correlated noise and atomic clock stabilization — that I have already read in several other papers. The work of the community in these topics is totally ignored. The paragraph about fidelity susceptibility has nothing to do with the rest of the manuscript. Also there relevant references are missing.

I agree that the technique is interesting and valuable if one is specifically interested in calculating the metrological sensitivity in certain situations. However, apart pure technical aspects, there is nothing really new here, and I have not learned much reading this paper.

Objectively, the core of the paper (namely tensor network techniques for quantum metrology) is entirely reported in the supplementary material: this does not make much sense to me. The

supplementary material is meant for minor details to support a main text that should stand by its own feet. The main text here is just introduction + applications, and all the scientific novelty is fully included in the supplementary, the authors must agree with me that there is something wrong.

The impact and interest of this paper is certainly not that of Nature Communications (results are not novel and are not of broad interest in the community). I do not recommend the publication. I definitely think that the paper should be rewritten, the technical part should appear in the main text and be the core of the manuscript, and the manuscript should be sent for publication to a more specialized journal. I am not willing to consider a revised version of the manuscript.

Reviewer #3:

Remarks to the Author:

The paper considers the problem of looking for the maximum metrological precision in large quantum systems numerically.

There are analytical precision bounds for uncorrelated local noise given in Ref. 11. Let us assume that we intend to estimate a parameter θ using N particles. These important analytic results show that above a certain particle number, essentially, the precision scales with $\text{var } \theta \sim 1/N$, which corresponds to the shot noise scaling. Thus, the best scaling called the Heisenberg scaling ($\text{var } \theta \sim 1/N^2$) cannot be achieved and the advantage of using entangled states is much more limited than what we could think on when studying theoretical calculations with highly entangled states, such as the Greenberger-Horne-Zeilinger (GHZ) states.

The present manuscript considers correlated local noise, which is not covered by calculations mentioned above. In this case, the analytic bounds mentioned above are not valid. It is interesting to obtain the precision achieved by various setups for such noise numerically. This is important for any practical application of quantum metrology. We need some tools that can model the system and estimate the achievable precision. If it is found that the shot-noise scaling can be overcome for setups for relevant particle numbers, this would be an extremely good news for the entire field of quantum information science.

The paper presents a numerical method to look for the quantum state maximizing the metrological precision. In particular, the paper uses an iterative maximization algorithm to find the maximum of the quantum Fisher information. The quantum Fisher information is a quantity convex in the quantum state and it has to be maximized over a convex set. Then, the maximum will be taken on the boundary, and the optimization problem looks hopeless. Authors suggest a truly very creative and smart solution to this problem. They show that their iterative method finds the global maximum.

After proposing a method for optimizing the quantum Fisher information, the authors consider the problem of large Hilbert space. Direct modeling of the system is not possible apart from very small systems. In order to model large systems, relevant to practical situations, authors use matrix product states. They also use matrix product states developed for infinite system.

Authors demonstrate the usage of their method via a number of relevant applications. Authors use their method to model magnetic field sensing with locally correlated noise. They look for the best metrological performance. They model the system with finite matrix product states and also with infinite matrix product states, and find a good agreement. Authors also model Atomic clock stabilization. That is, they look for the optimal initial state as well as other parameters for the best performance. For that, they have to minimize the Allan variance. Authors also use their method for fidelity susceptibility calculations for many-body thermal states.

The paper considers a very timely, important problem. There is a large effort in quantum information science to find cases when quantum systems outperform classical ones. One of the best candidates for finding such an example is quantum metrology. Authors present a very general method to optimize the precision in quantum metrological systems with uncorrelated local noise, and present a toolbox immediately usable in any applications. With this toolbox, many setups can be modeled looking for a quantum advantage.

I believe that the paper is a very important work for quantum information science, on the level of the very successful and very highly cited paper Ref 11. (while present paper is not a continuation in any sense). I very strongly suggest to publish the work in Nature Communications.

Dear Editor,

We would like to thank the Referees for their reports. We see that the main issue raised by all the Referees is the fact that too much material has been delegated to the Methods section and the Supplementary Information part at the expense of the the main body of the paper. We agree that this makes the paper less accessible to the reader and that without going to the Methods section the essential elements of the implementation of the MPO algorithm might not be clear. We must admit, that this situation was a result of earlier compromises we had to make in order to fit the requirements of the Nature Physics journal to which our manuscript was originally submitted and then transfer to Nature Communications. Since Nature Communications is more generous in this respect, there is indeed no reason not to include some more detailed explanations of the essential parts of the MPO implementation of the optimization algorithm. Therefore:

- We have included a whole new subsection entitled “Expressing the optimization problem using the MPO formalism” in the “Results” section of the main body of the manuscript. This subsection contains part of the material from the Methods section on expression of the states, their derivatives as well as the quantum evolution using the MPO formalism as well as some condensed description of the optimization steps of the algorithm.
- Since now the essential objects of the MPO description are described in the main body of the text we were able to expand a bit the discussion of the example of “Magnetic field sensing with locally correlated noise”, where we provide more details on the exact form of the evolution of the states in the MPO language.
- we have also made a number of amendments in response to the Editor’s queries, including slightly expanded version of the title, figure formatting and notational changes.

In order to ease the review process, we have highlighted all the changes with respect to the previous submission in the attached “TEXMetrology_diff.pdf” file.

Below we respond in details to all the Referees’ remarks. We quote the Referees’ comments in green, followed by our responses and the relevant changes in the manuscript in blue..

Referee #1

Comment 1.1

My main comment is with regard to the presentation. The paper only provides the main story while all the important details are either in the method section or the supplementary material. In my opinion some technical intuition should be added to the main text. 1) Some intuition regarding the validity of eq. 2 should be added when the equation is presented. 2) The fact that the derivative can be written as an MPO is a crucial part in the technique. This fact is mentioned in the main text but only explained in the method section. At least a short explanation should appear in the main text.

We hope that new arrangement of the paper satisfactory addresses Referee’s criticism. Indeed now the detailed description of how the derivative is written using the MPO formalism is fully described in the new subsection in the main body of the manuscript (fully addressing point 2) above).

Moreover concerning the point 1) we expanded a bit the discussion below Eq.2 by pointing out

that this is quadratic function for which the extremum condition is a linear equation equivalent to the SLD formula:

This form is equivalent to the standard definition of the QFI [31,32], as can be seen by solving the above maximization problem with respect to L —this is formally a quadratic function in matrix L and the resulting extremum condition yields the standard linear equation for the symmetric logarithmic derivative (SLD) L , $\rho'_\varphi = \frac{1}{2} (L\rho_\varphi + \rho_\varphi L)$. When the solution for L is plugged into the above formula it yields $F(\rho_\varphi) = \text{Tr}(\rho_\varphi L^2)$ in agreement with the standard definition of the QFI.

Comment 1.2

The section "Fidelity susceptibility calculations for many-body thermal states" shows that the Fisher Information is the susceptibility. An explanation could be added how come the Fisher Information does not depend on ϵ as the derivative with respect to the susceptibility does.

Both Fidelity susceptibility and Fisher information appear as coefficients in the expansion of the Fidelity in the second order in ϵ . Hence, strictly speaking neither of them depends on ϵ . To make it clear we added the following sentence highlighting the relation between the QFI and fidelity susceptibility:

QFI defines a metric in the space of quantum states (the Bures metric) [32] and is directly related with the fidelity susceptibility, namely $F = 4\chi_\varphi$.

This said, it is also true that when discretising the derivative we obtain these quantities up to some relative error. We explicitly state this below Eq.(26) where we write:

The accuracy of the fidelity susceptibility is limited by the finite bond dimension D_L as well the finite parameter difference ϵ . Nevertheless, we obtain satisfying results with relative error around 1% for $\epsilon = 10^{-4}$ and $D_L = 4$.

Comment 1.3

The authors mention on page 5 that "This shows that any analysis of the performance of atomic clocks based on the NOON state may be misleading." That fact is also known from many other investigations so should be probably erased or references should be added.

Indeed the Referee is correct, this is not a surprising observation and hence we have modified the sentence as follows:

The NOON states are highly prone to dephasing noise, and hence the optimal interrogation times will be necessary reduced compared to the optimal, more robust states. This is a manifestation of a generic poor performance of the NOON/GHZ states in realistic (noisy) scenarios with increasing particle numbers N [10,11,36].

Referee #2

Comment 2.1

The paper adapts known tensor network techniques to quantum metrology. The paper is quite technical and its ultimate goal is to compute the sensitivity of phase estimation protocols (namely the calculation of the quantum Fisher information) in presence of certain correlated noise (some additional assumptions are made). The applications are the standard ones magnetic field sensing with locally correlated noise and atomic clock stabilization that I have already read in several other papers. The work of the community in these topics is totally ignored. The paragraph about fidelity susceptibility has nothing to do with the rest of the manuscript. Also there relevant references are missing.

The critiques of the Referee is very general and hence it is not completely clear for us what particular results/references the Referee had in mind. Still, we will provide some additional arguments here in order to convince the Referee regarding the novelty and relevance of the results obtained.

First, it is true that we make use of the known tensor network techniques and adapt it to quantum metrology. Still, even though the term “adapt” might suggest that it is a straightforward application of the ready-to-use tools for some problems these tools had not been used before, this is very far from what actually the term adapt stands for in our case. The QFI that we use as a figure of merit has never been calculated before using the tensor network techniques in case of mixed states. When we started the project it was not clear for us that this project will succeed at all, as we were not apriori sure that it will possible to effectively approximate the SLD operator using a low bond dimension MPO and hence calculate QFI effectively. Moreover, it was essential to choose a proper formulation for the actual calculation of the QFI (see Eq. 2), as other equivalent forms where not suitable to be used within the MPO formalism. Our paper is therefore the first paper to show, that indeed the QFI figure of merit can be efficiently calculated using MPO operators in case of noisy metrological models where we necessarily deal with mixed states.

Second, we have indeed chosen examples which represent important classes of metrological models that appear in the literature. By no means, however, these models have been solved in the literature. The optimal protocols in magnetic field sensing models have only been solved in case of uncorrelated noise, and atomic clock stabilization problem has not been rigorously addressed in full generality with temporal correlations taken into account. In fact, shortly after putting the paper on the arxiv we were contacted by one of the leading groups in the field who has also studied the model of magnetic sensing problem with correlated noise, but focused their analysis solely on the performance of the GHZ type states [38], and did not make an attempt to find the performance of the optimal metrological protocol [38], which is what we were able to achieve using our approach. Similarly, in atomic clocks considerations, typically a number of ad-hoc assumptions is taken regarding the lack of correlations between subsequent local oscillator frequency estimates (see e.g [19]) and even though the arguments are often physically convincing, there is no rigorous optimization algorithm that would confirm that certain protocols are indeed optimal. Our approach provides new techniques to address these problems, and we have demonstrated their utility by calculating the Quantum Allan Variance quantity, in the regime inaccessible using the standard state representation.

Third, the purpose of the example with fidelity susceptibility calculations was to show that, abstracting from the task of optimizing quantum metrological protocols, the proposed method of

efficient calculations of the QFI may be of use in problems apparently unrelated with quantum metrology itself. In this respect the Referee is right that the third example is of different kind, but it is not fair to say that “it has nothing to do with the rest of the manuscript”—the fidelity susceptibility is directly related with the QFI and we calculate this quantity using the MPO methods developed in the paper.

Comment 2.2

Objectively, the core of the paper (namely tensor network techniques for quantum metrology) is entirely reported in the supplementary material: this does not make much sense to me. The supplementary material is meant for minor details to support a main text that should stand by its own feet. The main text here is just introduction + applications, and all the scientific novelty is fully included in the supplementary, the authors must agree with me that there is something wrong.

We are convinced that the newly reorganized version of the paper addresses this criticism in a satisfactory way.

We hope that the above explanations may prompt the Referee to take a more positive look at our results. We will also be willing to provide some more detailed explanations, if the Referee decides to make some points of his criticism more concrete.

Referee #3

There are analytical precision bounds for uncorrelated local noise given in Ref. 11. Let us assume that we intend to estimate a parameter θ using N particles. These important analytic results show that above a certain particle number, essentially, the precision scales with $\text{var } \theta \sim 1/N$, which corresponds to the shot noise scaling. [...]

The present manuscript considers correlated local noise, which is not covered by calculations mentioned above. In this case, the analytic bounds mentioned above are not valid. It is interesting to obtain the precision achieved by various setups for such noise numerically. This is important for any practical application of quantum metrology. [...]

In particular, the paper uses an iterative maximization algorithm to find the maximum of the quantum Fisher information. The quantum Fisher information is a quantity convex in the quantum state and it has to be maximized over a convex set. Then, the maximum will be taken on the boundary, and the optimization problem looks hopeless. Authors suggest a truly very creative and smart solution to this problem. [...]

The paper considers a very timely, important problem. There is a large effort in quantum information science to find cases when quantum systems outperform classical ones. One of the best candidates for finding such an example is quantum metrology. Authors present a very general method to optimize the precision in quantum metrological systems with uncorrelated local noise, and present a toolbox immediately usable in any applications. With this toolbox, many setups can be modeled looking for a quantum advantage. I believe that the paper is a very important work for quantum in-

formation science, on the level of the very successful and very highly cited paper Ref 11. (while present paper is not a continuation in any sense). I very strongly suggest to publish the work in Nature Communications.

We thank the Referee for his/her overall very positive comments and we hope that the he/she will agree that the amendments made in the paper only strengthen the message and improve the accessibility of the results presented. We also believe that the concrete arguments provided by the Referee 3 will will also be convincing for Referee 2 with respect to the novelty and importance of the results.

We once again thank all the Referees for their thoughtful remarks and hope that in the light of our replies and amendments to the manuscript, the Editor and the Referees will find the paper significant enough for publication in Nature Communications.

Yours faithfully,

The Authors